# Distinct Clinical and Laboratory Patterns of *Pneumocystis jirovecii* Pneumonia in Renal Transplant Recipients

**DOI:** 10.3390/jof7121072

**Published:** 2021-12-13

**Authors:** Andreas M. J. Meyer, Daniel Sidler, Cédric Hirzel, Hansjakob Furrer, Lukas Ebner, Alan A. Peters, Andreas Christe, Uyen Huynh-Do, Laura N. Walti, Spyridon Arampatzis

**Affiliations:** 1Department of Nephrology and Hypertension, Inselspital, Bern University Hospital, University of Bern, 3010 Bern, Switzerland; andreas.meyer.hagberg@gmail.com (A.M.J.M.); daniel.sidler@insel.ch (D.S.); uyen.huynh-do@insel.ch (U.H.-D.); spiros.arampatzis@hin.ch (S.A.); 2Department of Infectious Diseases, Inselspital, Bern University Hospital, University of Bern, 3010 Bern, Switzerland; cedric.hirzel@insel.ch (C.H.); hansjakob.furrer@insel.ch (H.F.); 3Department of Diagnostic, Interventional and Pediatric Radiology, Inselspital, Bern University Hospital, University of Bern, 3010 Bern, Switzerland; lukas.ebner@insel.ch (L.E.); alan.peters@insel.ch (A.A.P.); andreas.christe@insel.ch (A.C.)

**Keywords:** *Pneumocystis jirovecii* pneumonia, renal transplantation, infection, interhuman transmission

## Abstract

Late post-transplant *Pneumocystis jirovecii* pneumonia (PcP) has been reported in many renal transplant recipients (RTRs) centers using universal prophylaxis. Specific features of PcP compared to other respiratory infections in the same population are not well reported. We analyzed clinical, laboratory, administrative and radiological data of all confirmed PcP cases between January 2009 and December 2014. To identify factors specifically associated with PcP, we compared clinical and laboratory data of RTRs with non-PcP. Over the study period, 36 cases of PcP were identified. Respiratory distress was more frequent in PcP compared to non-PcP (tachypnea: 59%, 20/34 vs. 25%, 13/53, *p* = 0.0014; dyspnea: 70%, 23/33 vs. 44%, 24/55, *p* = 0.0181). In contrast, fever was less frequent in PcP compared to non-PcP pneumonia (35%, 11/31 vs. 76%, 42/55, *p =* 0.0002). In both cohorts, total lymphocyte count and serum sodium decreased, whereas lactate dehydrogenase (LDH) increased at diagnosis. Serum calcium increased in PcP and decreased in non-PcP. In most PcP cases (58%, 21/36), no formal indication for restart of PcP prophylaxis could be identified. Potential transmission encounters, suggestive of interhuman transmission, were found in 14/36, 39% of patients. Interhuman transmission seems to contribute importantly to PcP among RTRs. Hypercalcemia, but not elevated LDH, was associated with PcP when compared to non-PcP.

## 1. Introduction

In the context of universal prophylaxis, *Pneumocystis jirovecii* pneumonia (PcP) is now occurring late after solid organ transplantation; renal transplant recipients (RTRs) seem to be at increased risk [1,2,3,4,5,6,7,8,9,10,11].

With unspecific respiratory symptoms, the clinical manifestation of PcP is highly variable and may be insidious, making the diagnosis difficult, especially when occurring late after transplantation [12,13]. Common laboratory features associated with PcP include elevated serum lactate dehydrogenase (LDH), (1-3)-β-D-glucan and hypercalcemia [14,15,16,17,18].

In order to characterize possible distinct clinical and laboratory patterns when comparing PcP to non-PcP and to explore the incidence, time of onset and acquisition routes of the PcP occurring at our center, we conducted a retrospective single center cohort study.

## 2. Materials and Methods

Bern University Hospital (Inselspital) is a tertiary-care urban teaching hospital in Switzerland treating >40,000 inpatients and more than 500,000 outpatient consultations per year. The current study was a retrospective cohort analysis of confirmed cases of PcP (PcP cohort) and pneumonia with other presumably bacterial pathogens (non-PcP) from our local adult transplant recipient cohort, which is part of the Swiss transplant cohort study. The local renal transplant registries in our institution were primarily screened between 2009 and 2014 for patients with suspected PcP and secondary compared to RTRs with non-PcP between 2009 and 2018.

The standard immunosuppressive regimen at our center consists of triple therapy with calcineurin inhibitors (primarily ciclosporin or tacrolimus), antimetabolites (primarily mycophenolate mofetil) and steroids, which are tapered slowly. Our induction regime also includes basiliximab on days 0 and 2 after transplantation. PcP prophylaxis was usually given for 3–6 months at the start of the study. PcP diagnosis was based on consensus guidelines, requiring a positive direct microscopy by immunofluorescence on induced sputum or bronchoalveolar lavage (BAL) and/or a positive PCR assay on a BAL specimen [19]. In addition, we included RTRs with a high clinical and radiological suspicion of PcP and adequate clinical response to PcP treatment where induced sputum or BAL was not possible due to critical respiratory situation.

Non-PcP pneumonia was diagnosed according to common diagnostic criteria for bacterial pneumonia, based on radiologic evidence of a new, or progression of a previous, pulmonary infiltrate plus at least 2 of the following criteria: fever >38 °C, cough, purulent sputum, dyspnea or >20 breath/min, pleuritic chest pain and a leukocyte count of >10,000/mm^3^ or <4000/mm^3^ [20].

Data on the PcP prophylaxis, treatment, secondary prophylaxis, medication, laboratory parameters, dates of inpatient stays, visits to the outpatient department, underlying diseases, clinical condition and demographic data were retrospectively reviewed and extracted from the hospital information system.

Laboratory parameters (sodium, potassium, calcium, phosphate, C-reactive protein (CRP), creatinine, urea, glomerular filtration rate, LDH, leukocytes, lymphocytes, neutrophilic granulocytes, oxygen partial pressure (PO_2_), carbon dioxide partial pressure (PCO_2_) and pH) were analyzed three, two and one months before diagnosis as well as at admission. Laboratory parameters were compared between the two cohorts (PcP vs. non-PcP) during the course of disease at the same time points.

The local ethical committee has approved all research involving human participants (Ethics Commission of the Canton of Bern, Bern, Switzerland, No. 2017–01267). All clinical investigations were conducted according to the principles expressed in the Declaration of Helsinki.

For the statistical analysis, *p*-values < 0.05 were considered statistically significant. Laboratory parameters at the different time points within the PcP cohort or the non-PcP cohort, respectively, were compared using the Wilcoxon matched-pairs signed-rank test for nonparametric distribution to establish differences that might be associated with respective pneumonia. We also compared the laboratory parameters of the two cohorts at the same time points before diagnosis using the Mann-Whitney *U*-test. The demographics and clinical characteristics in Table 1 were compared using Mann-Whitney *U*-test or Z-test. Statistical analysis was performed with GraphPad Prism version 6.01 for Windows, GraphPad Software, La Jolla California USA, www.graphpad.com, 2012.

## 3. Results

Demographics and clinical characteristics of PcP and non-PcP in RTRs are presented in Table 1. 

During the study period, 36 PcP cases and 57 non-PcP cases were identified. Most patients were male and had received a cadaveric allograft.

In the PcP cohort, 34/36 (94%) patients were diagnosed with PcP by isolation of *P. jirovecii* from BAL specimens (79% (27/34) with immunofluorescence and 21% (7/34) by PCR). In the remaining two cases, clinical manifestations, computed tomography scan (CT scan) findings and resolution with trimethoprim/sulfamethoxazole (TMP/SMX) were highly suggestive for PcP. The duration from symptom onset to diagnosis was similar in both cohorts (10 days (interquartile range (IQR) 5–14) for PcP and 7 days (IQR 2–14) for non-PcP).

Based on the PcP cohort, with cases occurring from 2009 to 2014, the yearly incidence of PcP during this 6-year period ranged from 2.6 (2013) to 20.9 (2010) per 1000 patient years at risk, whereas the yearly incidence of non-PcP pneumonia was more constant (Figure 1). The cumulative incidence was 4.0% for PcP and 3.2% for non-PcP.

The median time between transplantation and pneumonia varied widely in both cohorts and was shorter for PcP (27 months (IQR 11–69)) than for non-PcP (86 months (IQR 15–132), *p* = 0.0401).

In the PcP cohort, late-onset PcP (>6 months after renal transplantation (RTx)) occurred in 30/36 cases (83%), thereof 50% occurred later than 22 months after transplantation.

In all 6/36 patients who developed early PcP, adequate prophylaxis had not been in place due to allergy (3/6), transient allograft dysfunction (1/6) and patient adherence issues (2/6).

Figure 2 illustrates occurrence of PcP over time. In 20/36 (56%) patients, PcP occurred over a period of 22 months (01/2009–11/2010); the remaining 16/36 (44%) cases occurred over a period of 46 months (03/2011–12/2014).

Encounters between RTRs diagnosed with PcP at the outpatient clinic were found for 14/36 (39%) patients, suggestive of a cluster. The median time between possible transmission event and PcP diagnosis was 34 (IQR 19–54) days.

One-third of PcP patients (13/36, 36%) had undergone renal biopsy with intensified steroid treatment thereafter within the previous 12 months, four of them (4/36, 11%) within three months prior to PcP diagnosis (Figure 2). In the last 3 months before PcP diagnosis, corticosteroid doses >20 mg/d for more than four weeks were given in 8/36 patients (22%) of recipients with PcP. Cytomegalovirus (CMV) reactivation was detected in 8/36 (22%) patients. Considering all these factors, no established individual risk factor to restart prophylaxis was found in 21/36 (58%) of the recipients experiencing PcP.

Clinically, cough was frequent in both cohorts (PcP: 72%, 23/32; non-PcP: 91%, 50/55; *p* = 0.0208) and dyspnea was predominantly existent in the PcP cohort (PcP: 70%, 23/33; non-PcP: 44%, 24/55; *p* = 0.0181). Fever was present in only one third of PcP cases but in three quarters of non-PcP patients (PcP: 35%, 11/31; non-PcP: 76%, 42/55; *p* = 0.0002).

Relevant laboratory values are shown in Figure 3. In the PcP cohort, sodium (*p* values < 0.01) and lymphocyte count (*p* values < 0.01) showed a decrease toward diagnosis at all time points (minus 3/2/1 month up to diagnosis). Calcium (*p* = 0.0006) and LDH (*p* values < 0.05) showed a significant increase before definite diagnosis. In the non-PcP cohort, only sodium (*p* = 0.0297) was found to be decreased at diagnosis. While the trend of the laboratory parameters was similar in both cohorts, serum calcium decreased in the non-PcP patients, in contrast to the PcP cohort.

Intensive care unit (ICU) admission was similar in both (PcP: 5/36, 14% vs. non-PcP: 10/57, 18%). Two patients (2/36, 6%) died during hospitalization for PcP, but death was not considered to be caused by the infection.

In the non-PcP RTRs cohort, 3/45 (7%) died during hospitalization, one due to respiratory failure.

## 4. Discussion

In a tertiary Swiss transplant center, we studied 36 consecutive RTRs with PcP and 54 with non-PcP. We aimed to characterize distinct clinical and laboratory patterns and prognosis of PcP compared to non-PcP cases and analyzed incidence, time of onset and factors associated with this cluster of PcP. We found an increased incidence of PcP during the second-year post-transplantation. Respiratory distress seemed more pronounced in PcP and fever less common. Nevertheless, outcomes of PcP and non-PcP were similar. PcP seems to be less associated with individual predictive factors but rather health care contacts with potential inter-individual transmissions. Hypercalcemia was frequently present at PcP diagnosis, whereas hyponatremia and increase in LDH were found in PcP and non-PcP.

Our findings regarding the time point of PcP occurrence after RTx are in agreement with recent studies. We found an increased incidence of late-onset PcP (>80% cases later than 6 months after transplantation) in our RTR cohort, similar to the reports of other centers [4,8,21,22,23]. In the PcP cohort, the median time from transplantation to PcP onset was 27 months. Other studies reported periods from 19 to 32 months in patients receiving 6 months of prophylaxis after transplantation [8,9]. As recommended in the current guidelines, a 3–6-month routine prophylaxis for PcP has shown to prevent the infection early after transplant very efficiently. The time of highest PcP-risk post-transplantation shifted beyond the first year post-transplantation [11].

Clinical presentation was unspecific in both cohorts. Respiratory distress was more pronounced in PcP compared to non-PcP, potentially reflecting the bilateral diffuse alveolar damage [11,24]. Interestingly, fever was only present in around one third of PcP cases, lower than historically reported but in line with recent larger studies exploring clinical symptoms of PcP pneumonias in RTRs [8,9,25,26]. Unspecific and atypical symptoms can contribute to the clinical challenge of distinguishing PcP from other pathologies involving the respiratory tract, including non-infectious causes [11].

Laboratory features traditionally associated with PcP, such as elevated LDH and decrease in serum sodium, were non-specific in our study population. Elevated LDH potentially reflects cellular damage occurring in severe pneumonia overall, rather than specifically in PcP. This challenges the reported good predictive values of LDH for PcP in RTRs [27].

Decrease in serum sodium has been described in other lower respiratory infections (e.g., Legionnaire’s disease). Traditionally, it has been assumed to be the result of increased production of antidiuretic hormone (ADH), which has been challenged recently [28].

Increased serum calcium levels during the course of PcP was the only laboratory alteration that differed between the two cohorts. We observed an increase in serum calcium in the PcP cohort that was not present in the non-PcP cases. This finding has been previously described in PcP, interestingly, mainly in RTRs [26]. Moreover in a large PcP cohort in France, hypercalcemia at PcP diagnosis was associated with less fever [26]. While this mechanism is unclear, increased serum calcium could be related to elevated 1,25-dihydroxyvitamin D levels in PcP [11,17]. Lymphopenia has been described as an increasingly relevant risk factor for late-onset PcP infection [4,8,9,11]. We observed a decrease in lymphocytes in both the PcP and non-PcP cohort, suggesting a general, pathogen-unspecific increased risk of infection occurring in these RTRs.

The clinical course was favorable in both cohorts, with similar in-hospital mortality.

In around half of the PcP cases, no reported individual risk factor (CMV infection, treatment of graft rejection, higher-dose corticosteroid therapy) indicating restart of PcP prophylaxis was identified [4,8,9,11,27].

Possible encounters between patients with PcP and other transplant recipients later diagnosed with PcP occurred within the cluster, suggesting that patient-to-patient transmission may be an important factor in the late onset of PcP, consistent with various reported outbreaks at other centers [3,5,6,7,11,29,30,31,32]. We were not able to prove inter-human transmission; moreover, transmission route is not clearly established. Droplet precaution in the sense of a general mask wearing when PcP cases are detected in a transplant cohort should be considered, as previously suggested [32].

To our knowledge, this is the first study comparing clinical and laboratory changes in PcP compared to non-PcP in a cohort of RTRs. Nevertheless, our study has several limitations. Foremost, its retrospective and observational nature and single center design limits the generalizability of our results. In addition, our outbreak includes two patients (6%) with only clinical PcP diagnosis, where diagnosis was not affirmatively possible (no BAL or induced sputum available) due to the critical respiratory status, reflecting real life challenges in the diagnosis of PcP. The design of our study limited the assessment of specific risk factors for PcP/non-PcP. Furthermore, we are not able to provide genotyping of *P. jirovecii* to support patient-to-patient transmission.

During the cluster outbreak in our center, primary prophylaxis was prolonged to 6 months and then to 12 months to protect the most susceptible patients. These measures have succeeded in reducing the number of new PcP infections. Thereafter, blanket prophylaxis was established in our patients in order to prevent further PcP cases.

## 5. Conclusions

This is the first study comparing clinical and laboratory changes in PcP compared to non-PcP in a cohort of RTRs. In our cohort, PcP occurred late after transplantation and the clinical course was favorable in both cohorts. Known individual risk factors indicating the restart of PcP prophylaxis were absent in a majority of recipients but encounters between PcP patients seemed to contribute importantly to the occurrence of late-onset PcP, suggesting a patient-to-patient transmission route. Hypercalcemia was the most distinct laboratory alteration observed at PcP diagnosis when compared to non-PcP cases. However, laboratory markers are not able to clearly distinguish PcP from non-PcP. Moreover, the classical approach to diagnosis, including awareness even late after transplantation as well as radiological and clinical picture and microbiological diagnosis, remains valid.

## Figures and Tables

**Figure 1 jof-07-01072-f001:**
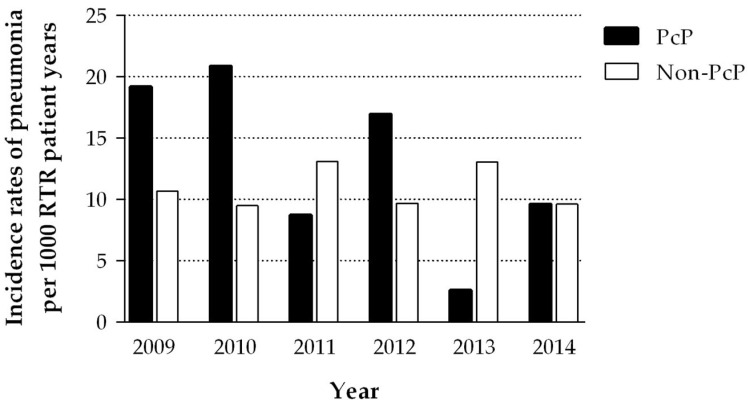
Incidence rates of pneumonia in renal transplant recipients (RTRs) at our center per 1000 patient years; black: *Pneumocystis jirovecii* pneumonia (PcP); white: pneumonia with other presumably bacterial pathogens (non-PcP). PcP shows an alternating incidence with particularly high incidence rates per 1000 patient years in 2009, 2010 and 2012, whereas the non-PcP cohort shows a relatively constant incidence.

**Figure 2 jof-07-01072-f002:**
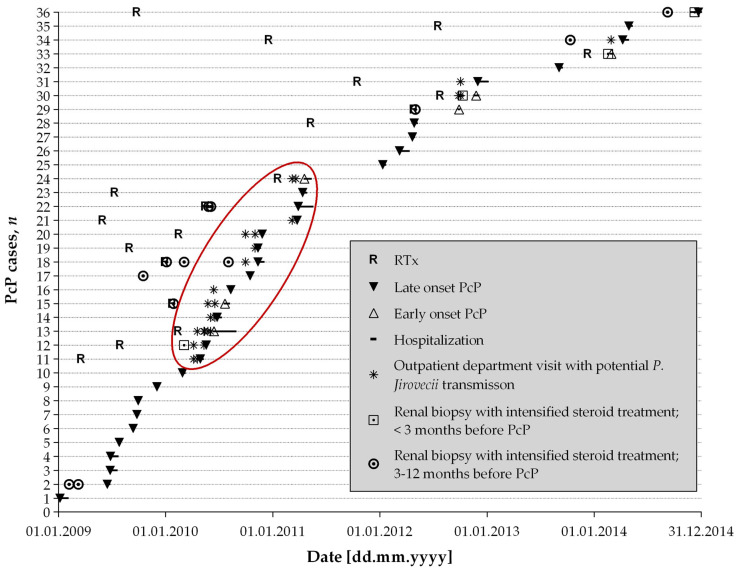
Map of 36 *Pneumocystis jirovecii* pneumonia (PcP) cases. RTx: renal transplantation. The *x*-axis reflects the timeline and the *y*-axis corresponds to the 36 individual cases sorted by time of PcP diagnosis. The red circle marks a potential PcP cluster with closely sequenced PcP cases and potential transmission events with concurrent visits to the outpatient clinic.

**Figure 3 jof-07-01072-f003:**
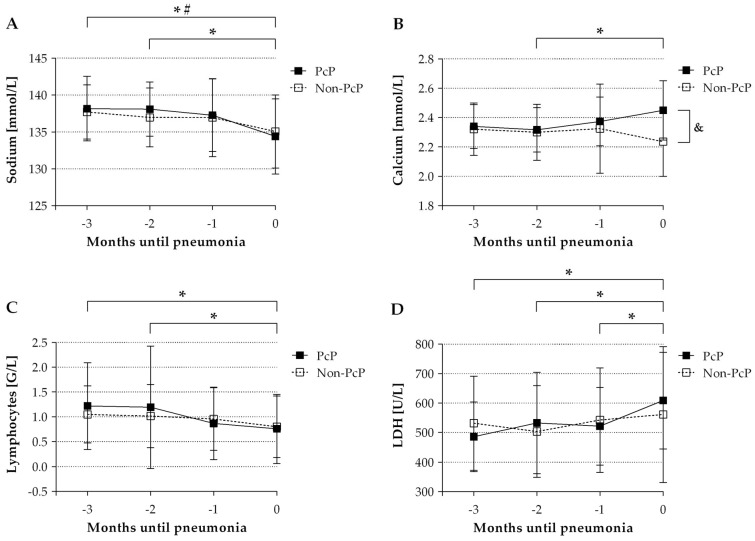
Course of sodium (**A**), calcium (**B**), lymphocyte count (**C**) and lactate dehydrogenase (LDH) (**D**) in renal transplant recipients (RTRs) with *Pneumocystis jirovecii* pneumonia (PcP) or pneumonia with other presumably bacterial pathogens (non-PcP). Analyses were made at the time of three, two and one months before diagnosis as well as at admission. Laboratory parameters were compared between the two cohorts at the same time points and within each cohort at different time points, respectively. * *p* < 0.05 for laboratory parameter in the PcP cohort at corresponding time points (Wilcoxon matched-pairs signed-rank); # *p* < 0.05 for laboratory parameter in the non-PcP cohort at corresponding time points (Wilcoxon matched-pairs signed-rank); & *p* < 0.05 for laboratory parameter in the PcP and non-PcP cohort at the same time point (Mann-Whitney *U*-test).

**Table 1 jof-07-01072-t001:** Demographics and clinical characteristics of PcP and non-PcP in renal transplant recipients (RTRs).

	PcP Cohort			Non-PcP Cohort			*p*-Value
	Median (IQR)	*N* = 36	(%)	Median (IQR)	*N* = 57	(%)	
Age (years)	58 (22–77)			58 (48–65)			0.7253
Male		26/36	72		38/57	67	0.5791
Cadaveric renal transplant		29/36	81		45/53	85	0.5968
Time between RTx and diagnosis (months)	27 (11–69)			85 (15–132)			0.0401 *
Symptoms at diagnosis							
➢ Fever		11/31	35		42/55	76	0.0002 *
➢ Cough		23/32	72		50/55	91	0.0208 *
➢ Dyspnea		23/33	70		24/55	44	0.0181 *
➢ Tachypnea (>20/min)		20/34	59		13/53	25	0.0014 *
➢ Pleuritic chest pain		2/33	6		8/53	15	0.2084
Time between symptom onset and diagnosis (days)	10 (5–14)			7 (2–14)			0.1710
Inpatient treatment		36/36	100		45/57	79	0.0033 *
Length of hospital stay (days)	12 (7–24)			8 (5–16)			0.0522
ICU Treatment		5/36	14		10/57	18	0.6473
➢ Length of ICU stay (days)	5 (3–5)			3 (1–5)			0.2970
Death within 30 days		2/36	6		3/45	7	0.8395
➢ Sepsis and ICH ➢ Fulminant CMV infection ➢ ARDS (H1N1) ➢ NSTEMI ➢ Pulmonary embolism					1/45	2	
					1/45	2	
					1/45	2	
		1/36	3				
		1/36	3				

PcP: *Pneumocystis jirovecii* pneumonia; non-PcP: pneumonia with pathogen other than *Pneumocystis jirovecii*; IQR: interquartile range; RTx: renal transplantation; ICU: intensive care unit; ICH: intracerebral hemorrhage; CMV: Cytomegalovirus; ARDS: acute respiratory distress syndrome; H1N1: Influenza A virus subtype H1N1; NSTEMI: non-ST-segment elevation myocardial infarction. * = significant (*p* < 0.05) difference between the PcP and non-PcP cohort (Mann-Whitney *U*-test or *Z*-test).

## Data Availability

Researchers interested in collaborating with our institution and accessing data are welcome to submit their proposal to the corresponding author and to the Ethics Committee of Bern following the rules laid out in the Swissethics BASEC guidelines and using the templates provided online https://swissethics.ch/en/basec (accessed on 20 October 2021).

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
