# Peer review of "Distinct Clinical and Laboratory Patterns of Pneumocystis jirovecii Pneumonia in Renal Transplant Recipients"

_jof, 2021, doi:10.3390/jof7121072_

Round 1

Reviewer 1 Report

This interesting work attempts to differentiate the clinical presentation of Pneumocystis pneumonia (PcP) (preferred as written. See doi: 10.2214/AJR.05.5158 for explanation). 

The manuscript focus and organization can both be improved.

The first 11 references are about clusters regardless that transmission is not the focus of the manuscript. Genotyping is lacking and therefore transmission or potential infective sources are conjectural at this stage.  

A few tips that hope might be useful:

Abstract: It takes the reader 3 sentences to reach the main purpose of the manuscript. In addition, potential encounters disctract the main focus of the manuscript. They could be commented in the discussion, but not necesarily need to appear in the abstract.  Especially since typing was not done.

Introduction: The first paragraph may be moved elsewhere or removed. 

Methods: Flow and organization need improvement. Were other etiologies diagnosed? The ethics statement is in the middle of the methods. 

Please add the work by Phipps et al to the references already cited, when discussing potential encounters (doi: 10.1097/TP.0b013e3182384b57).

The main strength of the manuscript is the clinical and laboratory data. Case definition could be strengthed if only the 27 microscopically confirmed plus the 7 confirmed by PCR (I understand that is single round PCR) PcP cases are compared with cases of pneumonia from other etiologies. Therefore, I suggest to remove the 2 non confirmed cases, and eventually consider a different category like possible cases.   "possible cases" will need to be defined as well. Other non-PcP etiologies should be listed when available.

Table 1: Statistics (P. value) should be added in a new column to the right.

Figure 2. This transmission map is difficult to interpret without genotyping, because whether the index and subsequent cases relate to a same genotype is unproven. I agree that distribution suggests index cases followed by a group of subsequent cases.  Consider re-doing the figure, and to place it in the mansucript after more clear-cut data. Cite Phipps, LM et al. Your manuscript would be so much stronger including genotyping. 

Please indicate the cause of death in the two  PcP, and in 3 non-PcP patients. 

Discussion: This section of the manuscript needs organization. The most complete data is the timing and clinical and laboratory data comparison of PcP versus non-PcP cases. 

Data on clustering  is weak and should be clearly commented as such and after the stronger data. 

Reviewer 2 Report

The authors present a retrospective observational single-center study of the clinical of the distinctive features of Pneumocystis jirovecii pneumonia (PJP) and non-PJP pneumonia in patients who have undergone renal transplant.

The authors describe their methodology and use appropriate statistic to to determine the distinctive features.

The manuscript is well written.

The authors should please explain why the reviewed data of renal transplant patients  from 2009 to 2018 (line 53) but limited their study to 2009 to 2014

Reviewer 3 Report

The paper describes PJP incidence in a retrospective Swiss cohort enriched for renal transplant recipients.  The paper provides some useful clinical insights.  However the statistical analysis appears fairly basic and on wonders if more meaningful insights could be gleaned from the dataset. 

  1. Table I should have statistical method added to the legend as well as a column for p Values.
  2. Is there a relationship between CMV reactivation and PJP
  3. Given the late onset of PJP, the authors should conduct cumulative drug analyses or area under the curve analysis to see if cumulative drug exposure is a risk of PJP.
  4. Do the authors have any data on CD4/CD8 ratio?
  5. Any discussion of outbreak or transmission would clearly require genotyping. The authors should clearly state that and avoid making any conclusions without molecular evidence. 
  6. Statistical methods should be added to the legends.

Round 2

Reviewer 1 Report

The responses to this reviewer comments are adequate and the beautiful clinical work is clear now.

Minor changes are needed. Please read a more recent article about Pneumocystis nomenclature (https://doi.org/10.1093/mmy/myab024), and do the following changes:

Title: delete "jirovecii".

Abstract: delete jirovecii and add P. jirovecii from the first sentence as follows:

Reads: Pneumocystis jirovecii pneumonia (PcP) has been reported in many...

Should read: Pneumocystis pneumonia (PcP) has been  reported in many...

I reccomend to keep jirovecii in the abstract for searching purposes, and so, would suggest to modify the next sentence as follows:

Reads: Specific features of PcP compared to...

Should read: Specific features of this fungal pneumonia caused by P. jirovecii compared to...

Line 79: Typo ... clinical response to PcJP... 

The authors may want to consider to report differences on chest radiographies in a future manuscript.